nanotechnology/environmental chemistry/environmental engineering

$Ga_2Zr_{2-x}W_xO_7$, fluorite phase, wastewater treatment, nanomaterials, crystal violet dye, photocatalytic degradation

**Author for correspondence:**
Rabab A. Nasr
e-mail: rababelsheikh@yahoo.com

This article has been edited by the Royal Society of Chemistry, including the commissioning, peer review process and editorial aspects up to the point of acceptance.

# Decolourization of crystal violet using nano-sized novel fluorite structure $Ga_2Zr_{2-x}W_xO_7$ photocatalyst under visible light irradiation

H. A. Abbas[1], Rabab A. Nasr[2], Rund Abu-Zurayk[3], Abeer Al Bawab[3] and Tarek S. Jamil[2]

[1]Inorganic Chemistry Department, and [2]Water Pollution Control Department, National Research Center, El Behouth Street, PO Box 12622, Dokki, Cairo, Egypt
[3]Chemistry Department School of Science, The University of Jordan, Hamdi Mango Centre for Scientific Research, 11942, Amman, Jordan

HAA, 0000-0002-6387-224X; RAN, 0000-0003-2613-7032; RA-Z, 0000-0002-1072-1624; AAB, 0000-0003-2131-1791; TSJ, 0000-0002-1856-8046

Fluorite-type Zr-based oxides with the composition $Ga_2Zr_{2-x}W_xO_7$ ($x = 0$, 0.05, 0.1, 0.15 and 0.2) were prepared using the citrate technique. Appropriate characterizations of all prepared materials were carried out. X-ray diffraction clarified that the undoped and W-doped $Ga_2Zr_2O_7$ samples were crystallized in the cubic fluorite phase structure. The average particle size of the samples was in the range of 3–8 nm. The lowest band gap (1.7 eV) and the highest surface area (124.3 $m^2\,g^{-1}$) were recorded for $Ga_2Zr_{0.85}W_{0.15}O_7$. The photocatalytic impacts of the prepared systems were studied by removal of crystal violet (CV) dye employing visible light illumination and taking into consideration the initial dye concentrations, duration of visible irradiation treatment, catalysts dose and the dopant concentration. The obtained results showed higher dye removal with the boost of the catalyst dosage. W doping shifted the absorption to the visible light range by decreasing the band gap from 4.95 eV for parent $Ga_2Zr_2O_7$ to 1.7 eV for 15 mol% tungsten-doped $Ga_2Zr_2O_7$ enhancing the photocatalytic decolourization of CV from 4.2% to 83.6% for undoped and 15 mol% W-doped $Ga_2Zr_2O_7$, respectively, at optimum operating conditions (pH 9, 1 g $l^{-1}$ catalyst dose and 300 min) while heavily doped W sample containing 20 mol% W showed lower removal than 15 mol% W-doped $Ga_2Zr_2O_7$. Complete CV degradation using 15 mol% W-doped $Ga_2Zr_2O_7$ was attained with the assistance of 25 mmol $l^{-1}$ hydrogen peroxide. The reaction is aligned to

pseudo-first-order kinetics. Different scavengers were introduced to decide the significance of the reactive species in CV degradation. $O_2^{-\bullet}$ and $h^+$ had the major role in the degradation of CV by $Ga_2Zr_{2-x}W_xO_7$ system compared with $HO^\bullet$.

# 1. Introduction

Crystal violet (CV) dye is triphenylmethane cationic dye (figure 1). It is used in textile and paper dye industries as well as navy blue and black inks for printing, ball-point pens and inkjet printers. It is also used to colourize diverse products such as fertilizers, antifreeze, detergents and leather. CV is also used as a histological stain, particularly in Gram staining for classifying bacteria [1,2].

Disposal of dyes in wastewater is a source of water contamination and disturbance of aquatic life [3]. Therefore, a suitable and efficient method is critically required to treat the wastewater containing dyes such as CV [4,5] for its proven carcinogenic and mutagenic properties in animals [6,7] and in humans [8].

Conventional techniques such as biodegradation, coagulation, adsorption, physical deposition conventional oxidants and coagulants were inefficient for CV treatment [9,10]. On the other hand, advanced oxidation processes (AOPs) such as microwave catalysis, photocatalysis, membrane technique and advanced oxidants [11–13] are promising in CV decolourization.

The essential defect of physical treatment involves only moving the dyes from the liquid to the solid state which is not easy to decontaminate [14]. Therefore, chemical treatment using AOPs, especially the heterogeneous photocatalysis, received attention for degrading such pollutants [15]. In heterogeneous AOPs, the metal oxides produce some powerful non-selective hydroxyl radicals ($HO^\bullet$) that dissociate a wide range of organic contaminants [16] into short-chain aliphatic acid that is easier to be completely degraded [17]. In the UV–visible light, the electron–hole pair mechanism is demanded in order to introduce intermediate organic compounds that might be completely mineralized at the surface of metal oxides attaining green end products [17]. From the economic side, novel nano-sized photocatalysts' response to visible light received valuable consideration since it is cost-effective compared with UV light [18–20].

Decolourization of CV has been studied using different oxidants such as nanosphere $TiO_2$ [20], Mn-doped and PVP-capped ZnO NPs [21], Ag-modified Ti-doped-$Bi_2O_3$ [22], ZnS NPS [23], $CeO_2$–$TiO_2$ nanocomposite [24], AgBr–ZnO nanocomposite [25], grafted sodium alginate/ZnO/graphene oxide [26]. Afterwards, the performance of the prementioned oxides will be compared with the results of the present study.

$A_2B_2O_7$ oxides (where A and B abbreviate trivalent lanthanides elements and tetravalent D and F groups elements, respectively) have either a pyrochlore-type or a defect fluorite-type structure. They have attractive physical and chemical properties, such as high melting point, high thermal expansion coefficient, low thermal conductivity, high thermal stability, high radiation stability and high electrical conductivity. Consequently, they are used in several applications such as solid electrolytes, thermal barrier coating materials, nuclear waste host materials and high-temperature heating elements [27]. The electrical properties of the pyrochlores vary from highly insulating through semiconducting to metallic behaviour [28]. Many studies prepared various pyrochlore metal oxides such as $La_2Zr_2O_7$ (that acts as thermal barrier coating), [29] $Y_2Sn_2O_7$ (that acts as excellent host matrices for photoluminescence) [30] and $Gd_2Zr_2O_7$ (that acts as a proper host material for fixation of some of the nuclear waste products) [31]. Several pyrochlore-type oxides such as $K_2Ta_2O_6$ [32], $Na_2Ta_2O_6$ [33], $Pb_2Sn_2O_6$ [34], $KAl_{0.33}W_{1.67}O_6$ [35], Ag/Sn-doped $KSbTeO_6$ [36], Ag/Sn-doped $KSbTeO_6$ [37], $Na_2Ta_2O_6$ [38] and $ASbO_3$ [39] were used for the decolourization of dyes such as acid red G, Congo red, methyl orange, methylene blue and rhodamine B. The luminescence properties of $Ln_2Ce_2O_7$ fluorite-type is also studied [40]. There is a deficiency in the literature regarding the preparation of fluorite-type structure for various applications. Besides, the photocatalytic activity of $Ga_2Zr_{2-x}W_xO_7$ fluorite-type system for CV degradation has not been reported yet. In this frame, the present work aims to prepare and characterize nano-sized $Ga_2Zr_{2-x}W_xO_7$ fluorite-type system using a reliable, cost-effective, eco-friendly and easy method to optimize the shape and grain size of the nano-sized metal oxides (the Pechini method [41]). Tungsten as a dopant is selected due to the difference in the oxidation state and ionic radii of $W^{6+}$ and $Zr^{4+}$ that will permit studying the impact of doping on both the structural and photocatalytic activity of $Ga_2Zr_2O_7$. Furthermore, W is used for the reduction of the band gap in order to use $Ga_2Zr_{2-x}W_xO_7$ system in visible light irradiation. Finally, the photocatalytic activity has been studied for the prepared systems in CV dye removal in visible light and the reaction operation conditions were adopted (reaction time, pH, catalyst dose and initial pollutant concentrations).

**Figure 1.** Chemical structure of crystal violet.

## 2. Experimental

### 2.1. Preparation and characterization of the prepared materials

$Ga_2Zr_{2-x}W_xO_7$ system was prepared; where $x = 0$, 0.05, 0.1, 0.15 and 0.2 using the citrate technique (Pechini method) which is a wet-chemical method based on polymeric precursor [41] that was used to prepare several metal oxides [42–46].

In this method, α-hydroxy acid (citric acid) is used to chelate the cations forming a polybasic acid. Polyhydroxy alcohol (ethylene glycol) reacts with these chelates forming ester and water. Heating the mixture leads to polyesterification and after the evolution of nitrous oxide and water, the gel is obtained. The thermal decomposition of this gel results in a chemically homogeneous powder containing the desired stoichiometry [42,43].

Zirconium (IV) oxynitrate hydrate (Sigma-Aldrich), tungsten (VI) chloride (Sigma-Aldrich), gallium (III) nitrate (Silverton, San Diego), ethylene glycol (Sandycroft, Deeside, Clwyd) and citric acid anhydrous extra pure (LobaChemie) are used as starting materials. All chemicals were reagent grade and used as received without any modification.

$Ga_2Zr_2O_7$ was prepared using the Pechini method as follows: aqueous zirconium oxynitrate and gallium nitrate solutions were mixed, considering the desired stoichiometry of the metal oxides in the final ceramic powder solution (A). The citric acid (CA) was then added to the solution (A) to chelate metal cations at the CA : Me molar ratio of 4 : 1. Me denotes $Ga^{3+}$, $Zr^{4+}$ in the final ceramic powder. After dissolving the CA, ethylene glycol (EG) was added into the solution at a CA : EG molar ratio of 1 : 1.5. The solution was then heated at 140°C and kept under stirring to promote the esterification and polymerization reactions. After elimination of nitrous oxides and water, a gel was obtained. The gel was charred gradually up to 300°C then heated in the muffle furnace at 300°C for 2 h. The charred gel thus produced was ground and calcined for 2 h at 500°C, then ground and calcined for 2 h at 600°C. $Ga_2Zr_{2-x}W_xO_7$ systems where $x = 0.05$, 0.1, 0.15 and 0.2 were prepared using the same sequence. For the preparation of $Ga_2Zr_{2-x}W_xO_7$ samples, tungsten chloride was dissolved in ethanol and then added to the solution (A). Flowchart of the preparation of $Ga_2Zr_2O_7$ powder is presented in the electronic supplementary material, S.1. The samples' identification, as well as their composition, are presented in table 1.

X-ray diffraction (XRD) is the standard technique for determination of the crystal structure of a solid. XRD is used to identify the crystal structure, to determine the lattice parameters. The XRD measurements were carried out using 7000 Shimadzo (Japan) 2 kW model X-ray spectrophotometer with a nickel-filtered Cu radiation (CuKα) with $\lambda = 1.54056$ Å. The scanning $2\theta$ range was 5–80 with a step size of

**Table 1.** The composition determined by ICP compared to the expected composition of the prepared samples.

| sample | sample compositions | expected (wt%) | | | experimental (wt%) | | |
|---|---|---|---|---|---|---|---|
| | | Ga | Zr | W | Ga | Zr | W |
| ZG | $Ga_2Zr_2O_7$ | 32.14 | 42.05 | 0 | 32.26 | 42.13 | 0 |
| ZGW1 | $Ga_2Zr_{1.95}W_{0.05}O_7$ | 31.80 | 40.57 | 2.10 | 32.91 | 40.43 | 2.13 |
| ZGW2 | $Ga_2Zr_{1.9}W_{0.1}O_7$ | 31.48 | 39.11 | 4.15 | 31.39 | 39.18 | 4.18 |
| ZGW3 | $Ga_2Zr_{1.85}W_{0.15}O_7$ | 31.14 | 37.69 | 6.16 | 31.19 | 37.63 | 6.20 |
| ZGW4 | $Ga_2Zr_{1.80}W_{0.2}O_7$ | 30.82 | 36.30 | 8.13 | 30.77 | 36.37 | 8.15 |

0.2. The lattice parameters were determined using a program called UnitCellWin [47]. FTIR spectra were recorded in the frequency range 400–4000 cm$^{-1}$ with a resolution of 4 cm$^{-1}$ using FTIR 6100 Jasco (Japan) spectrum equipment.

Diffuse reflectance measurements were performed to study the optical properties of the prepared samples using Shimadzu UV-3600 (Japan). The free radicals created (EPR signals) were recorded at room temperature by X-band EMX spectrometer (Bruker, Germany) using a standard rectangular cavity of ER 4102 operating at 100 kHz field modulation. The microstructures were studied by transmission electron microscope (TEM, JEOL JEM2100, Japan). The specific surface area of the prepared samples was determined by NOVA surface area analyser from Thermo Pascal 140 mercury porosimetry under a pressure range of 0.1–200 MPa. Mercury surface tension of 480 dyne cm$^{-1}$ and the contact angle of 141.3° were used. Elemental analysis was carried out using inductive coupled plasma-atomic spectrometry (Agilent ICP-OES).

## 2.2. Photodegradation activity

CV was obtained from Sigma-Aldrich Chemical Company. All solutions were prepared in double-distilled water. Photocatalytic experiments were carried out with CV dye solution using all the prepared catalysts under visible irradiation. Irradiation was carried out by commercial visible metal halide lamp (HQI-T250/Daylight, OSRAM GmbH, Germany) with a luminous efficacy of 82 lm W$^{-1}$ and luminous flux of irradiation 20 000 lm.

A stirred slurry composed of dye solution and catalyst was placed in the dark for 30 min in order to establish equilibrium between adsorption and desorption phenomenon of dye molecule on the photocatalyst surface. Then the lamp was turned on and the slurry was magnetically stirred for homogeneous distribution of catalyst in the solution. At specific time intervals, an aliquot (5 ml) was collected and centrifuged for 2 min at 3500 r.p.m. to remove catalyst particles from aliquot to assess the extent of decolourization. The absorption spectra recorded 588 nm as $\lambda_{max}$ on the double-beam UV–visible spectrophotometer (Cary-100). The desired pH of the solution was adjusted by the addition of previously standardized 0.050 M $H_2SO_4$ and 1.0 M NaOH solutions. Performance efficiency was calculated as

$$\%\text{efficiency} = \frac{C_o - C}{C_o} \times 100, \tag{2.1}$$

where $C$ and $C_o$ are initial and final dye concentration, respectively, for reaction time $t$.

## 2.3. Evaluation of active species

To check the influence of some active species on the catalytic activity of catalyst trapping experiments were carried out for difference species. In these experiments, 1 mmol l$^{-1}$ of three scavengers were used which are isopropyl alcohol (IPA), ethylene diamine tetra acetic acid (EDTA) and benzoquinone for HO$^{\bullet}$, h$^{+}$ and O$^{2\bullet}$ species, respectively.

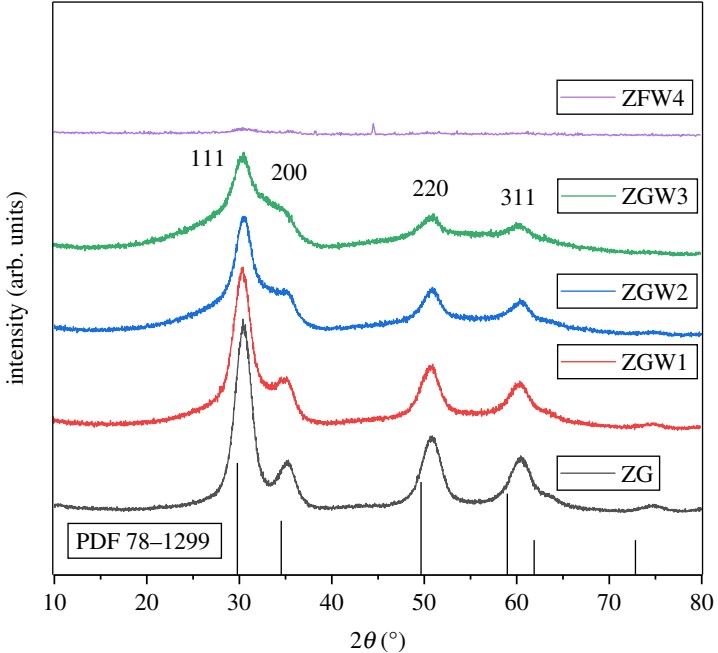

**Figure 2.** Powder X-ray diffraction pattern of $Ga_2Zr_{2-x}W_xO_7$ system calcined at 600°C/2 h ($x = 0$, 0.05, 0.1, 0.15 and 0.2).

# 3. Results and discussion

## 3.1. Characterization of the prepared materials

The crystallization of pyrochlore or fluorite phases for the mixed oxides $A_2B_2O_7$ depends on the radius ratio of A and B cations ($r_A/r_B$) in addition to the conditions of samples processing [48–50]. $A_2B_2O_7$ crystallizes in the stable pyrochlore structure when the $r_A/r_B$ is in the range of 1.46–1.78 [51] depending on the coordination number. The defect fluorite structure (cubic, Fm3m) is obtained for the lower or upper limits of the previously mentioned range of $r_A/r_B$. In the present study, the radius ratio $r_A/r_B$ for A = $Ga^{3+}$ (ionic radius = 62 pm) and B = $Zr^{4+}$ (ionic radius = 72 pm [52]) ions was found to be 0.85 which is lower than the above-mentioned range. In this frame, it is predicted that $Ga_2Zr_2O_7$ will be crystallized in the fluorite structure. This is confirmed by the XRD pattern of $Ga_2Zr_2O_7$ (ZG sample) calcined at 600°C for 2 h (figure 2) where ZG sample has the cubic fluorite phase structure (PDF 78–1299 for $Er_{0.5}Zr_{0.5}O_{1.75}$, which is the best-matched card that can be used where there is no card for the novel $Ga_2Zr_2O_7$ material). The shift to lower $2\theta$ value is due to the difference in the ionic radius between $Er^{3+}$ (ionic radius = 89 pm [52]) and $Ga^{3+}$ (ionic radius = 62 pm [52]) ions. The peaks at about 14°, 28°, 37°, 45° $2\theta$ corresponding to (1 1 1), (3 1 1), (3 3 1), (5 1 1) planes [53], respectively, characteristic for the pyrochlore structure, do not exist. Accordingly, ZGW1, ZGW2 and ZGW3 samples have the cubic fluorite phase structure (figure 2). Traces of cubic fluorite phase structure were detected for ZGW4 sample, which means that this sample may need further calcination in order to improve its crystallinity, which is not in alliance with the calcination temperature of all the prepared samples in the manuscript.

According to the ionic radius of $Zr^{4+}$ (ionic radius = 72 pm [52]) and $W^{5+}$ (ionic radius = 62 pm [52]) or $W^{6+}$ (ionic radius = 60 pm [52]) ions, it is predicted that the cubic lattice parameter and unit cell volume will decrease as the W concentration increases because ionic radius of $W^{5,6+}$ ion is less than that of $Zr^{4+}$ and also $Ga^{3+}$ ions. Surprisingly, it was found that as the W concentration raised, the cubic lattice parameter and unit cell volume are increased (table 2). This might be attributed to the substitution of $Zr^{4+}$ ions by $W^{5,6+}$ ions creating a distorted coordination environment and leading to unit cell expansion [54].

ICP was used to determine the chemical composition of the prepared materials by determining the wt% of Zr, Ga and W in all the prepared samples. The experimental wt% of Zr, Ga and W in all the prepared samples were in alliance with those of the expected wt% (table 1) indicating that the prepared materials have the exact proposed chemical compositions shown for ZG, ZGW1, ZGW2, ZGW3 and ZGW4.

The FTIR spectra of the prepared samples are shown in figure 3. For the parent ZG sample, the broad absorption peak at about 3422 $cm^{-1}$ is due to the stretching vibration of OH group in water molecule. The absorption band at about 1634 $cm^{-1}$ is characteristic to the bending vibration of the water molecules [55]. The band at about 460 $cm^{-1}$ is due to Zr–O vibration [56]. The peaks at 1040 and 1080 $cm^{-1}$ might be

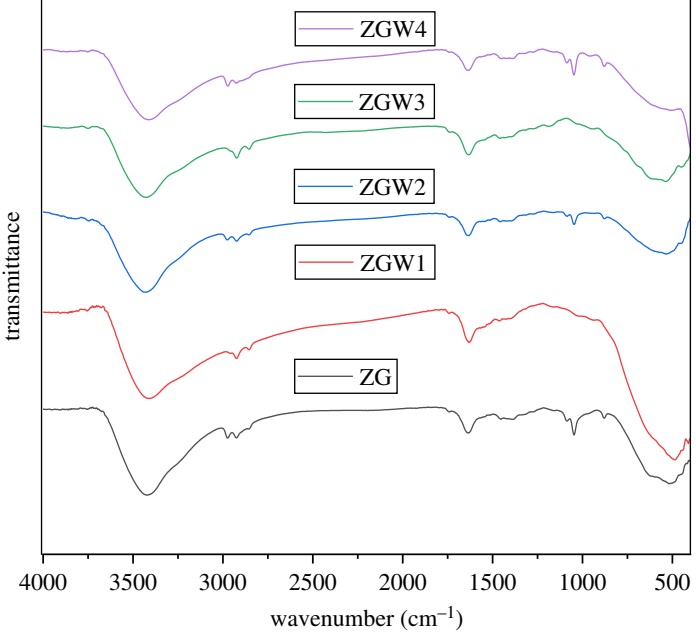

**Figure 3.** FTIR spectra of $Ga_2Zr_{2-x}W_xO_7$ system where $x = 0$, 0.05, 0.1, 0.15 and 0.2.

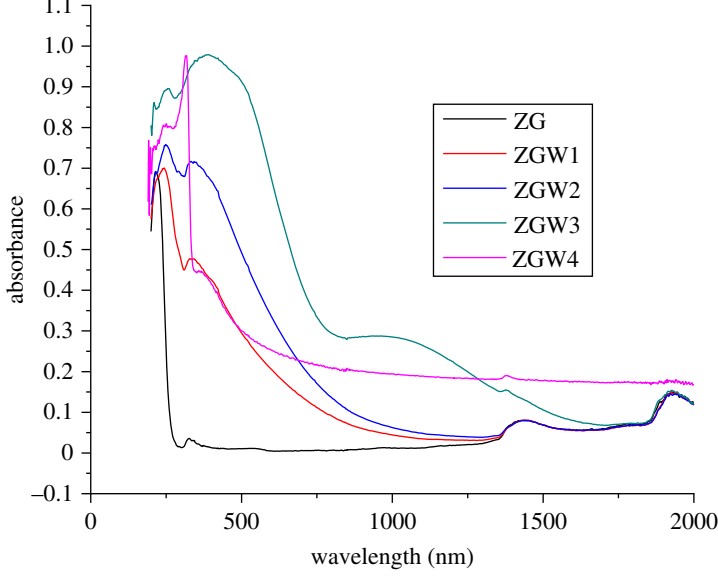

**Figure 4.** The diffuse reflectance spectra of ZG, ZGW1, ZGW2, ZGW3 and ZGW4 samples.

associated with stretching vibrations of Zr–O terminals [57]. The peak at about 880 cm$^{-1}$ corresponds to the bending vibration of hydroxyl groups bounds to zirconium oxide [55]. The bands at about 510 and 620 cm$^{-1}$ are assigned to Ga–O stretching and Ga–O–Ga torsion movements [58]. For the doped samples (ZGW1, ZGW2, ZGW3 and ZGW4), a little shift in the peak positions and peak intensities were observed with the increasing in the W concentration. The region 600–900 cm$^{-1}$ might correspond to O–W–O stretching modes [59]. The peak at about 620 cm$^{-1}$ is assigned to W–O$_{inter}$–W bridging vibration of the corner-sharing WO6 octahedron [60].

Diffuse reflectance spectroscopy (DRS) was used to determine the system optical properties. The absorption edge of ZG sample is about 247 nm. As shown in figure 4, W doping (till 15 mol%, ZGW3 sample) shifted the absorption to the visible light range (red shift). For ZGW4 sample ($Ga_2Zr_{0.18}W_{0.2}O_7$) the tungsten doping shifted the absorption to lower wavelength. The band gap is a very important parameter for the photocatalytic performance of the photocatalyst where it indicates the range at which the photocatalyst will be active (UV–visible light ranges).

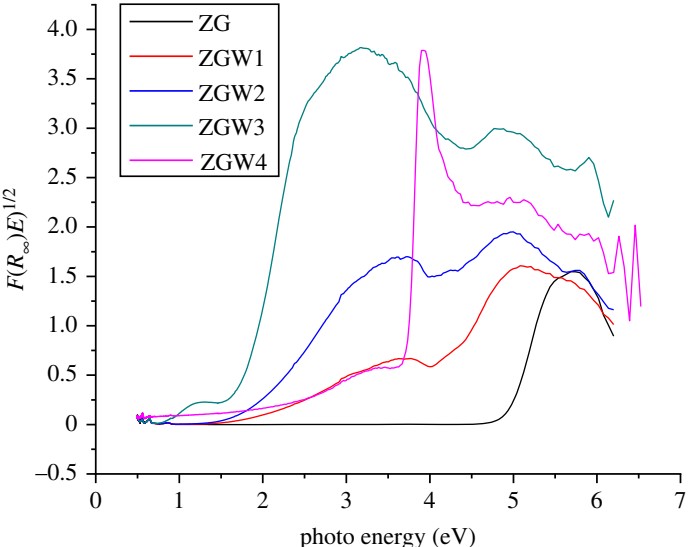

**Figure 5.** The plots of $F(R_\infty)E)^{1/2}$ versus photo energy for the estimation of the band gap energy for ZG, ZGW1, ZGW2, ZGW3 and ZGW4 samples.

**Table 2.** Microstructural parameters, the estimated band gap from DRS, the TEM particle size range and surface area for $Ga_2Zr_{2-x}W_xO_7$ system.

| sample | $a$ (Å) | $V$ (Å3) | TEM particle size range (nm) | band gap (eV) | surface area $m^2 g^{-1}$ |
|---|---|---|---|---|---|
| ZG | 5.08693 | 131.6337 | 4–7 | 4.95 | 91.3 |
| ZGW1 | 5.08706 | 131.6437 | 4–8 | 3.88 | 96.5 |
| ZGW2 | 5.08824 | 131.7358 | 3–5 | 1.81 | 99.9 |
| ZGW3 | 5.09405 | 132.1876 | 3–4 | 1.7 | 124.3 |
| ZGW4 | — | — | — | 2.66 | 90 |

The Kubelka–Munk function $F(R)$ is correlated to the diffuse reflectance $R$ according to the following equation:

$$F(R) = \frac{(1 - R^2)}{2R}. \tag{3.1}$$

Figure 5 shows the plots of $F(R_\infty)E)^{1/2}$ versus photo energy for the estimation of the band gap energy for ZG, ZGW1, ZGW2, ZGW3 and ZGW4 samples. The band gap value was determined by drawing $(F(R).h\upsilon)^{1/2}$ against photo energy and extrapolating the linear part of the curve to $(F(R).h\upsilon)^{1/2} = 0$ according to Kubelka–Munk using linear fit [61] (electronic supplementary material, S.2).

Table 2 shows that the band gap of ZG is 4.98 eV. The band gap is decreased from 3.88 to 1.7 by increasing the W concentration from 5 to 15 mol% while at 20 mol% W, the band gap increased again to be 2.66 eV.

For the understudied novel $Ga_2Zr_{2-x}W_xO_7$ system, the band gap of $Ga_2Zr_2O_7$ (4.95 eV) is closer to that of $ZrO_2$ (5 eV) [62] which renders its absorption of visible light accordingly, it is predicted to have a limited photocatalytic activity under visible light. The band gaps of ZGW2 and ZGW3 samples are smaller than that of $ZrO_2$ [63], $Sm_2Zr_2O_7$ (2.86 eV) [63] and $Nd_2Zr_2O_7$ (2.67 eV) [63]. The band gap of ZGW4 sample is similar to that of $Nd_2Zr_2O_7$.

For $Ga_2Zr_{2-x}W_xO_7$ system in this study, the conduction band (CB) is composed of Zr 4d orbitals whereas the valence band (VB) consists of the O 2p bands.

Electron paramagnetic resonance (EPR) is one of the most important tools used to describe the defects in solids because most of the defects contain unpaired electrons. Intrinsic or extrinsic point defects exist in solids. The intrinsic defects are present in the solid itself without introducing any impurity; when

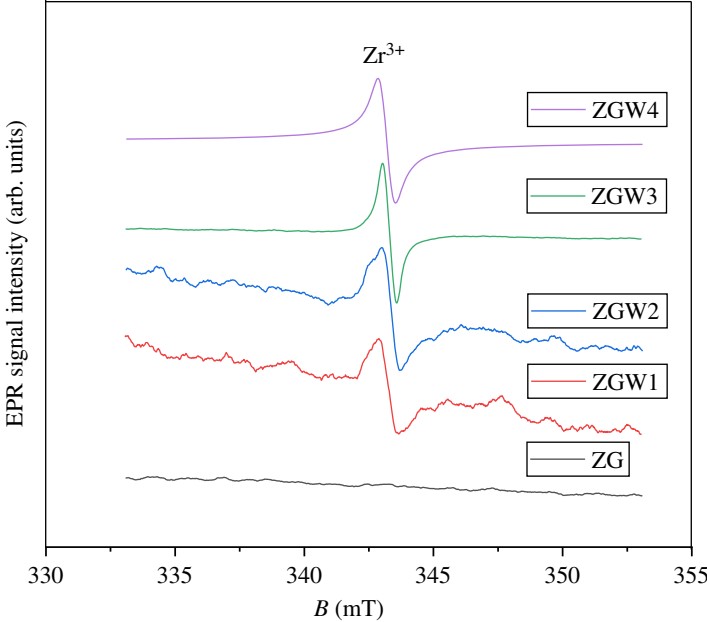

**Figure 6.** EPR spectra for ZG, ZGW1, ZGW2, ZGW3 and ZGW4 samples.

**Table 3.** The spin numbers for the prepared samples.

| sample | spin numbers |
|---|---|
| ZG | 0 |
| ZGW1 | $4.83 \times 10^{16}$ |
| ZGW2 | $4.33 \times 10^{17}$ |
| ZGW3 | $1.89 \times 10^{18}$ |
| ZGW4 | $1.25 \times 10^{17}$ |

chemical impurity is introduced, the extrinsic defects are produced. Recently, the impurities (dopants) have been used in photocatalysis to modulate the optical properties to improve the photocatalytic performance of the catalyst [64]. EPR spectra for the prepared samples are presented in figure 6, where no EPR spectra are detected for the undoped $Ga_2Zr_2O_7$ (ZG) sample. For W-doped samples (ZGW1, ZGW2, ZGW3 and ZGW4 samples), an EPR signal is detected. For W-doped samples, $Zr^{4+}$ ion is substituted by $W^{5+,6+}$ ion which results in the creation of free electrons for electroneutrality, these electrons are trapped to $Zr^{4+}$ ion forming $Zr^{3+}$ ion which corresponds to the detected EPR signal [65]. No signals were detected for the oxygen vacancy. Table 3 shows the spin number (free radicals) for the prepared samples. As the W concentration increases, the spin number increases up to 15 mol% W (ZGW3 sample) and for 20 mol% sample (ZGW4 sample) the spin number decreases which is in accordance with DRS results. DRS results demonstrated that, $Zr^{3+}$ point defect might introduce a new energy level between the CB and the VB resulting in decreasing the band gap with increasing the W concentration up to 15 mol% W (ZGW3 sample). Increasing the band gap from 1.7 eV for ZFW3 sample to 2.66 eV for heavily doped ZGW4 sample might be due to the donor electrons of W filling the lowest level of the CB (the Burstein–Moss effect) [66]. The schematic energy level diagram for the prepared $Ga_2Zr_{2-x}W_xO_7$ system and its corresponding charge separation towards dye degradation under visible light illumination is presented in electronic supplementary material, S.3.

The TEM micrographs of samples are presented in the electronic supplementary material, S.4, S.5. Small quasi-spherical particles, which agglomerate into denser aggregates were observed. The TEM particle size range for all samples is presented in table 2. The lattice fringes with an interplanar distance of 0.25 nm which could be assigned to (2 0 0) plane were detected for the cubic phase of $Ga_2Zr_2O_7$ for ZGW3 sample (electronic supplementary material, S.6).

To manifest the surface area of the prepared oxides, the Barrett–Joyner–Halenda (BJH) nitrogen adsorption tests were used (figure 7). A type-IV isotherm demonstrates the typical mesopore materials

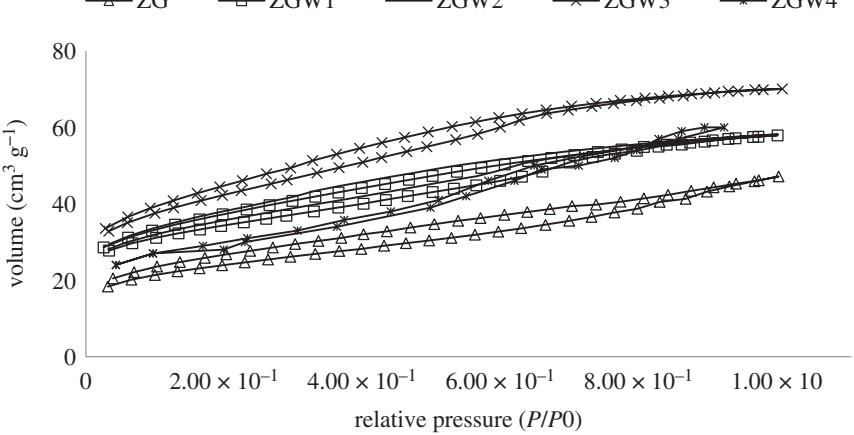

**Figure 7.** $N_2$ adsorption and desorption isotherms for prepared materials sample.

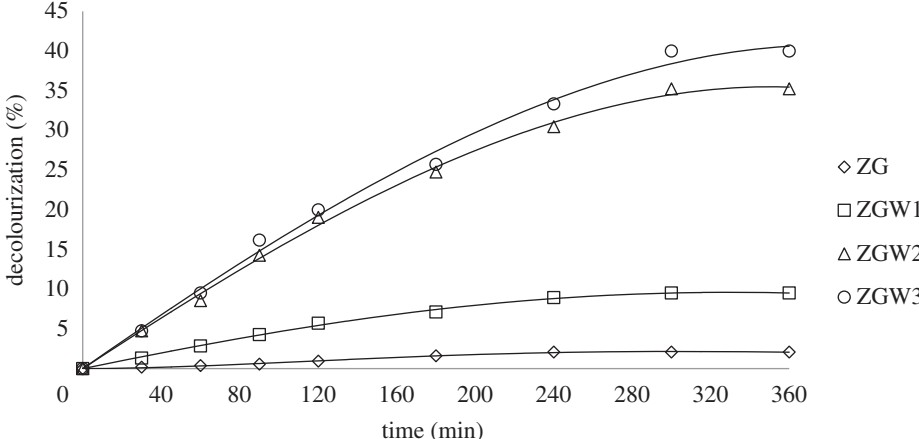

**Figure 8.** Photocatalytic degradation of 10 mg $l^{-1}$ CV under visible light by 0.75 g $l^{-1}$ of all the prepared materials as a function of time at pH 7.

that are related to aggregates presented in mesopores, and the little-marked uptake over a range of high $P/P_o$ [61,67]. Specific surface areas were significantly increased with the increment of tungsten concentration in the order of 91.3, 96.51, 99.9 and 124.3 $m^2\,g^{-1}$ for ZG, ZGW1, ZGW2 and ZGW3 samples, respectively, and decreased to be 90 $m^2\,g^{-1}$ for ZGW4 sample.

## 3.2. Photocatalytic activity of $Ga_2Zr_{2-x}W_xO_7$ system

### 3.2.1. Degradation time influence

Figure 8 demonstrates the influence of reaction time on photodegradation of CV by using $Ga_2Zr_{2-x}W_xO_7$ system under visible irradiation for 6 h. The degradation rate was increased with the increment of time till 5 h then it became stable after that, concluding that 5 h is the optimum reaction time. The decolourization % of CV dye after 5 h was recorded 2.1%, 9.5%, 35.23% and 40% for ZG, ZGW1, ZGW2 and ZGW3, respectively, at initial CV concentration of 10 mg $l^{-1}$ at pH 7 and 0.75 g $l^{-1}$ catalyst dose. Concluding that the order of photocatalytic activity was in accordance with the reduction in their band gaps (table 2).

Moreover, $O_2$ and HO groups on the surface were converted to $O_2^{-\bullet}$ and $HO^{\bullet}$, respectively. Both of them can assist in CV degradation in the following equations:

$$Ga_2Zr_{2-x}W_xO_7 + h\nu(\lambda > 400 \text{ nm}) \rightarrow Ga_2Zr_{2-x}W_xO_7(e_{cb-} + h_{vb}), \tag{3.2}$$

$$Ga_2Zr_{2-x}W_xO_7(e_{cb-}) + O_2 \rightarrow Ga_2Zr_{2-x}W_xO_{7+}O_2^{-\bullet}, \tag{3.3}$$

$$h^+ + H_2O \rightarrow HO^{\bullet}, \tag{3.4}$$

$$e^- + O_2 \rightarrow O_2^{-\bullet}, \tag{3.5}$$

$$h_{vb} + \text{dye} \rightarrow \text{degradation by-product} \tag{3.6}$$

and
$$O_2^- + \text{dye} \rightarrow \text{degradation by-product}. \tag{3.7}$$

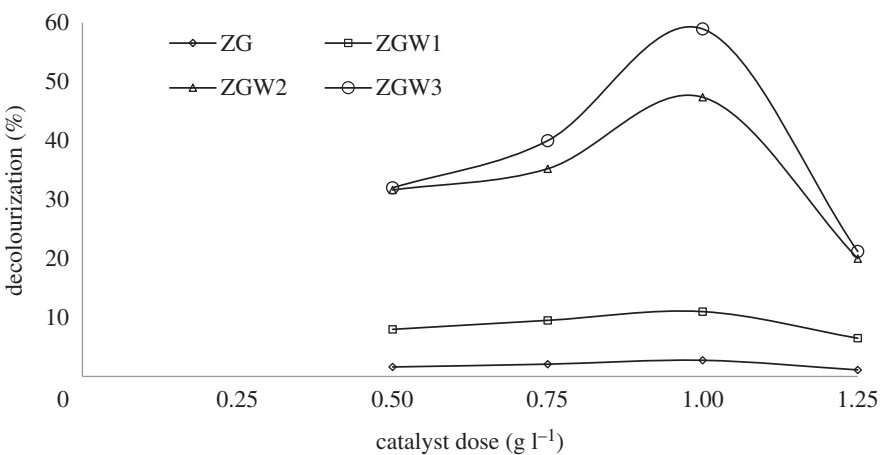

**Figure 9.** Photocatalytic decolourization under visible light for all the prepared materials as a function of catalysts dose (CV concentration 10 mg l$^{-1}$ at pH 7).

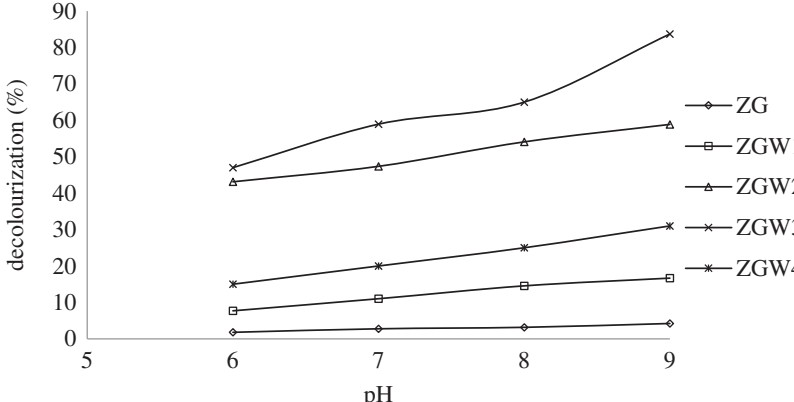

**Figure 10.** Photocatalytic degradation under visible light for all the prepared materials as a function of pH (CV concentration 10 mg l$^{-1}$ at 1 g l$^{-1}$ catalyst dose).

### 3.2.2. Catalyst load impact

The degradation of 10 mg l$^{-1}$ CV at optimum reaction time (5 h) and neutral pH were studied at different catalysts loads (0.5–1.25 g l$^{-1}$).

Figure 9 displays the direct relation between the % decolourization and the catalyst's loads for both undoped and W-doped GZ catalysts up to 1 g l$^{-1}$ due to intense numbers of catalytically active sites by increasing of catalysts load [68], that raises the rate of $O_2^{-\bullet}$ and $HO^{\bullet}$ creation [19]. Reduction of decolourization rate after 1 g l$^{-1}$ was noted and was owed to the excess catalyst amount impedes the light penetration [20,68,69]. Accordingly, 1 g l$^{-1}$ was elected as the optimum dose for CV degradation.

### 3.2.3. Influence of pH

Different pH values (6–9) under pre-optimized time and catalyst load were employed to decolourize 10 mg l$^{-1}$ CV. The % decolourization was directly proportional to pH value up to pH 9 with % decolourization of 4.2%, 16.6%, 58.88%, 83.7% and 31.2% for GZ, GZW1, GZW2, GZW3 and ZGW4, respectively (figure 10). In acidic medium, very low photodegradations of CV were spotted due to its hard deposition on the catalyst surface [70–73]. On the other hand, in alkaline medium, the amount of hydroxide ions increased and their availability to be converted to $HO^{\bullet}$ increased leading to an acceleration of the degradation rate [72].

Additionally, the impact of pH 10–pH 12 on dye decolourization were examined but not taken into consideration because at pH > 9, a colourless CV molecule occurred without illumination [73]. Consequently, pH 9 was elected to be the optimum pH.

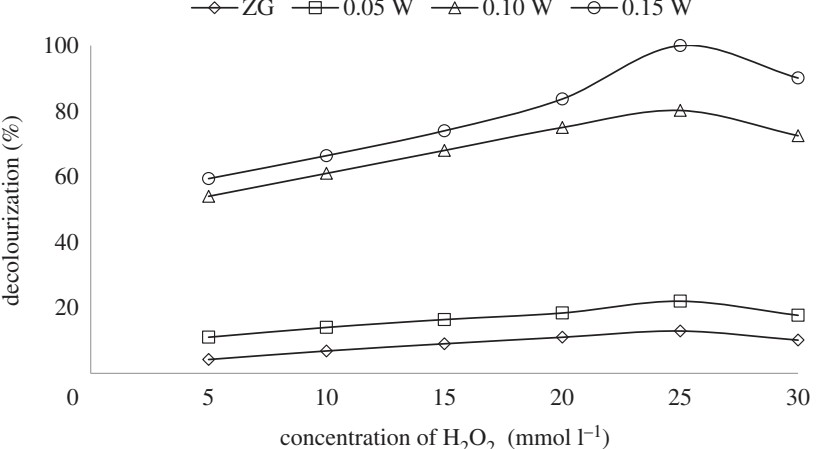

**Figure 11.** The effect of $H_2O_2$ doses on % decolourization of CV for ZGW3 (CV concentration 10 mg $l^{-1}$ at 1 g $l^{-1}$ catalyst dose and pH 9).

### 3.2.4. Effect of doping on the photocatalytic performance

The effect of W doping was explained in several parts in the manuscript including XRD, DRS, EPR as well as the photocatalytic efficiency. The positive role of doping for certain concentration might be attributed to (i) formation of new energy levels between the VB and the CB; these levels act as effective charge carrier traps, and (ii) improving the adsorption of the molecules of the pollutant on the surface of the catalyst by altering the catalyst surface acid–base properties [74]. For the prepared samples, the photocatalytic degradation of CV increases as W concentration increases, as shown in figure 10, which is in accordance with their band gap values in the DRS part. ZGW3 sample has high CV removal compared with the undoped ZG sample. This might be attributed to the substitution of $Zr^{4+}$ by $W^{5+,6+}$ in the $Ga_2Zr_2O_7$ lattice, which is reflected in the cubic lattice parameter and unit cell volume values for ZG and ZGW3 sample and introduction of new $Zr^{3+}$ level between the VB and the CB of $Ga_2Zr_2O_7$ (electronic supplementary material, S.3), decreasing the band gap as well as increasing the spin numbers detected by EPR. After visible light illumination, the electron is promoted from the VB to the CB through the $Zr^{3+}$ level and the hole is formed. The electron and hole react with the adsorbed oxygen and hydrogen peroxide forming $O_2^{-\bullet}$ and $HO^\bullet$ which are responsible for CV photodegradation [75]. For the highly W-doped sample (ZGW4), the decrease of the photocatalytic degradation of CV as compared with the other W-doped samples might be due to filling of the lowest level of CB by W donor electrons [66] which is matched with the band gap values calculated by DRS.

### 3.2.5. Effect of $H_2O_2$

Hydrogen peroxide is counted as one of the most fundamental photo-oxidants in water decontamination, hydrogen peroxide is used in the visible radiation to assess $HO^\bullet$ radicals generation, which is the essential promoter for the destruction of toxic organic compounds as represented in the following equation [76]:

$$H_2O_2 + \text{visible irradiation} \rightarrow 2HO^\bullet. \tag{3.8}$$

$H_2O_2$ doses of 0–30 mmol $l^{-1}$ were used to study the effect of $H_2O_2$ on % decolourization of 10 ppm CV at pH 9 by 1 g $l^{-1}$ catalyst as presented in figure 11. Increasing $H_2O_2$ dose from 0 to 25 mmol $l^{-1}$ is directly proportional to the % CV degradation attaining complete degradation for ZGW3 after 5 h due to the increase in the amount of $HO^\bullet$ promoting the degradation rate. Further addition of $H_2O_2$ to 30 mmol $l^{-1}$ decreased the % decolourization of CV due to hydroxyl radical and hole scavenging effects (equation (3.9)) [77]. Hence the optimum dose of $H_2O_2$ was elected as 25 mmol $l^{-1}$ [78,79].

$$H_2O_2 + HO^\bullet \rightarrow HO^{2\bullet} + H_2O. \tag{3.9}$$

### 3.2.6. Influence of initial dye concentration

The influence of different initial CV concentrations was demonstrated under prementioned optimum conditions with ZGW3. As affirmed in figure 12 which was set up on the linear relationship, the

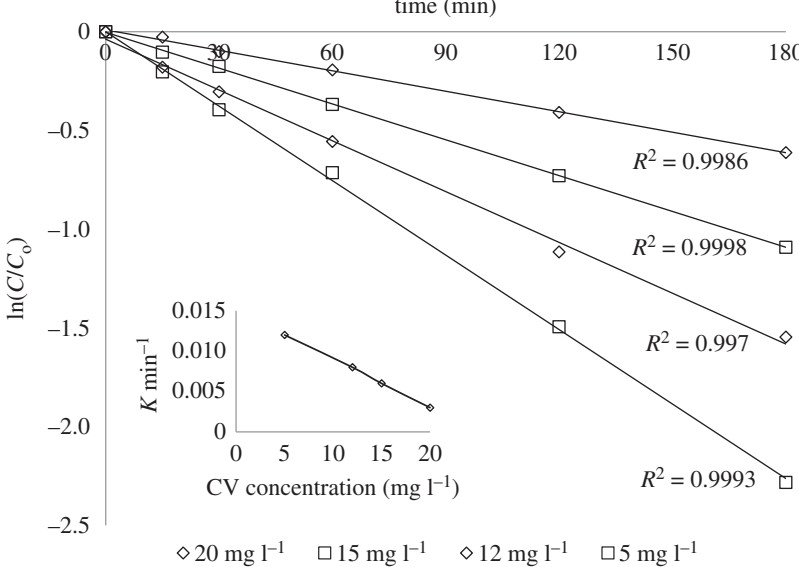

**Figure 12.** Pseudo-first-order kinetics for different CV doses for ZGW3 (CV concentration 10 mg l⁻¹ at 1 g l⁻¹ catalyst dose and pH 9). Inset: the effect of different CV doses on the rate of the degradation reaction for ZGW3.

decolourization of CV arraigned by the pseudo-order kinetics in the following equation:

$$\ln\frac{C}{C_o} = K_{app}, \tag{3.10}$$

where $K_{app}$ is the spotted rate constant, $C_o$ and $C$ are the concentrations of CV at zero time and at a certain time, respectively.

Using ZGW3, the photocatalytic rate has increased with decreasing dye concentrations in the tested solutions which attained 0.012, 0.008, 0.006 and 0.003 min⁻¹ at CV concentration 5, 12, 15 and 20 mg l⁻¹, respectively (figure 12). For the adequately low initial CV concentration (5 mg l⁻¹), the dye was completely degraded after the photocatalytic time of 3 h. Additionally, increment in initial CV concentrations, photocatalytic degradation time increased to attain the nearly complete decolourization of CV (figure 12). This might be credited to the augmentation of optical densities of the CV dye solutions with the increment of dye concentrations, which may act as a filter to the incident light [72] and consecutive possible restrict of irradiation penetration to the catalysts' surfaces in all of the test solutions. In this manner only, fewer photons can arrive at the catalyst surface, and therefore the creation of HO⁰ free radical on the surface of the catalyst declined, since the available effective sites of catalyst become covered by the crowded dye ions. This results in decolourization rate reduction [68,72,73,76,77,80,81].

## 3.2.7. The degradation pathway of crystal violet

UV–visible spectra of CV dye solution as a function of reaction time for ZGW3 are depicted in figure 13. As noted from these spectra, at 0 time of the experiment prior to the oxidation reaction, the absorption spectrum of CV in water was distinguished by one main peak in the visible region ($\lambda = 584$ nm) and by two other peaks in the UV region ($\lambda = 250$ and 300 nm). The peaks at 250 and 300 nm were related to aromatic structures in the molecule, and that at 584 nm originated from the chromophore [82]. The gradual decay of the visible peaks with time was owing to the cleavage of the aromatic rings by oxidation. In addition to this rapid decolourization effect, the decrease of the absorbance at 250 or 300 nm was considered as an index of aromatic fragment degradation of the dye molecule and its intermediates [83,84].

GC–MS study was conducted to further recognition of the intermediate products formed at the end of the photocatalytic reaction figure 14. Based on the results and previous studies [82–85], figure 15 proposed initial degradation pathways that start with N-de-methylation followed by an attack of the oxidizing species on the central carbon portion of the CV to form 4-(N,N-dimethylamino)-4′-(N′.N′-dimethylamino) benzophenone [82,85]. Then the central carbon was successively attacked by the active radicals [85]. Finally, the gradual cleavage of the aromatic intermediates would lead up to the

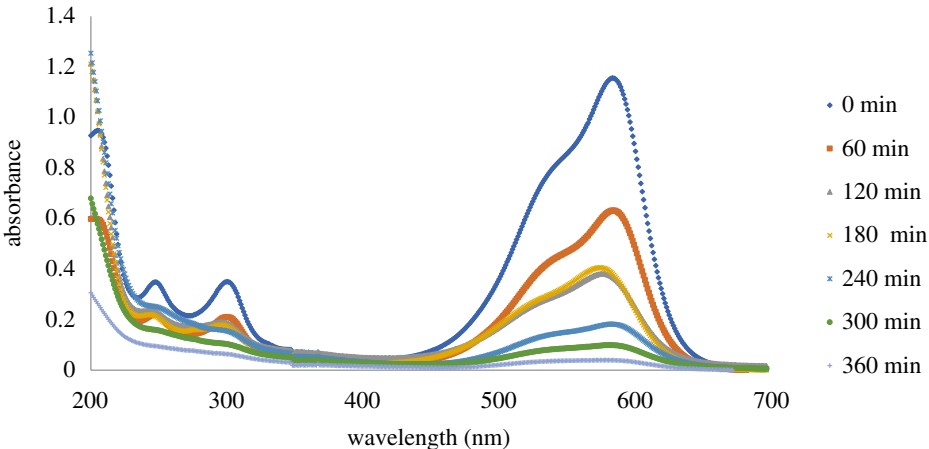

**Figure 13.** UV–visible spectra study of CV dye as a function of reaction time for ZGW3 (CV concentration 10 mg l$^{-1}$ at 1 g l$^{-1}$ catalyst dose and pH 9).

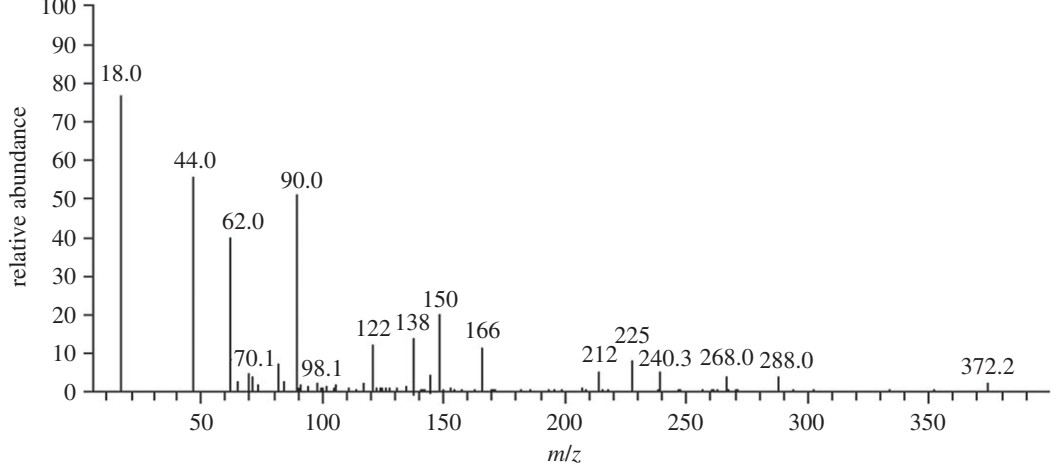

**Figure 14.** The GC–MS mass spectra of photocatalytic degradation for CV using ZGW3 under optimum operating condition (CV concentration 10 mg l$^{-1}$ at 1 g l$^{-1}$ catalyst dose and pH 9).

formation of carboxylic acids before transformation into carbon dioxide and water. The treated water was safe to be used for water remediation since it was non-toxic for *Vibrio fischeri* organism according to the test performed by Microtox analyser 500 [86].

The Photocatalytic efficiency of the prepared $Ga_2Zr_{2-x}W_xO_7$ for CV dye degradation in comparison with that of various other photocatalysts is presented in table 4. The prepared nano-sized cubic fluorite $Ga_2Zr_{1.85}W_{0.15}O_7$ oxide showed promising photocatalytic activity for decolourization of the harmful CV dye under visible light irradiation (which is more applicable from an economic view) compared with modified $TiO_2$.

### 3.2.8. The reusability for ZGW3 sample

The reusability of the catalyst is one of the main obstacles to the application of photocatalyst in water treatment. In order to examine the reusability, 10 cycles for CV decolonization over ZGW3 sample were accomplished under the pre-optimized operating conditions. The catalyst was deposited settling the solution for enough time. After detaching the supernatant, the catalyst had been introduced for another cycle. The variation in % CV removals with various cycles is presented in figure 16. The trivial diminishment in photocatalytic adequacy (100–94%) pointed to satisfactory results obtained with increasing the number of runs up to 10 affirming that the prementioned sample can be reused without losing the profitable synergy activity.

$C_{25}H_{30}N_3$

372.9

$C_{18}H_{19}N_3$

288

$C_{17}H_{20}N_2O$

268

$C_{13}H_{12}N_2O$

212

$C_{16}H_{20}N_2$

240

**Figure 15.** The proposed pathway for photocatalytic degradation for CV.

$C_{15}H_{15}NO$

225

$C_9H_{10}O_2$

150

$C_8H_6O_4$

166

$C_7H_6O_2$

122

$C_7H_6O_3$

138

$C_2H_6O_2$

62

$C_2H_6O_2$

62

$C_3H_6O_3$

90

$C_3H_6O_3$

90

**Figure 15.** (*Continued.*)

### 3.2.9. Evaluation of active species

Free radicals trapping experiments were conducted for ZGW3, as the highest catalytic activity sample, to explore the significant contributor in the photodegradation reaction under optimum operating condition (CV concentration 10 mg l$^{-1}$ at 1 g l$^{-1}$ catalyst dose and pH 9). When 1 mmol l$^{-1}$ IPA was introduced as

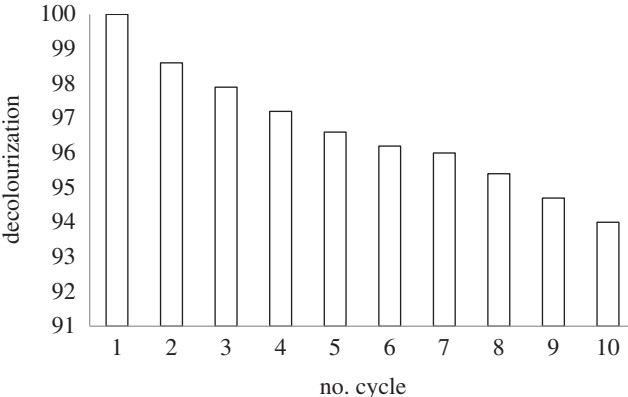

**Figure 16.** Number of cycles for CV over ZGW3 sample under visible light at optimum conditions (CV concentration 10 mg l$^{-1}$ at 1 g l$^{-1}$ catalyst dose and pH 9).

**Table 4.** Comparison of photocatalysts used for the degradation of CV dye.

| catalyst | efficiency % | irradiation source | time | cited by |
|---|---|---|---|---|
| BiOCl/H$_2$O$_2$ | 100 | visible | 8 h | [19] |
| anatase nanosphere TiO$_2$ | 99 | UV | 6 h | [20] |
| Mn-doped and PVP-capped ZnO NPs | 100 | UV–visible irradiation | 3 h | [21] |
| Ag-modified Ti-doped-Bi$_2$O$_3$ | 65 | UV | 90 min | [22] |
| TG-capped ZnS NPs | 87 | UV–visible irradiation | 3 h | [23] |
| AgBr–ZnO/H$_2$O$_2$ nanocomposite | 86.93 | visible light | 50 min | [25] |
| grafted sodium alginate/ZnO/graphene oxide | 94 | sun light | 5 h | [26] |
| Ga$_2$Zr$_{2-x}$W$_x$O$_7$/H$_2$O$_2$ | 100 | visible light | 5 h | present work |

HO$^\bullet$ scavenger, the degradation of CV was not clearly influenced (CV % removal accomplished 80%), which demonstrated that the HO$^\bullet$ was not the major reactive species. However, the degradation process could be hindered proficiently when 1 mmol l$^{-1}$ benzoquinone (BQ) was added, since CV removal was decreased to 45%. It indicated that the O$_2^{-\bullet}$ played a demonstrating role in the catalysis process. When 1 mmol l$^{-1}$ EDTA was added, the catalytic degradation of CV could be extremely inhibited (CV removal accomplished to 21%), indicating the h$^+$ also played a major role in the catalytic process. Consequently, h$^+$ and O$_2^{-\bullet}$ had the major contribution to photocatalytic degradation of CV while, HO$^\bullet$ had a minor contribution [87].

## 4. Conclusion

Nano-sized Ga$_2$Zr$_{2-x}$W$_x$O$_7$ system is prepared successfully in the cubic fluorite phase using the Pechini method where $x = 0$, 0.05, 0.1, 0.15 and 0.2. XRD, IR, EPR, XPS, TEM, BET, ICP and diffuse reflectance are used for the characterization of the prepared samples. The undoped in addition to W-doped Ga$_2$Zr$_2$O$_7$ has cubic fluorite phase structure. According to XRD, it was found that the samples are in the nano-sized range (3–4 nm). The band gap of the Ga$_2$Zr$_2$O$_7$ (4.95 eV) is close to that of ZrO$_2$ (5 eV). W doping decreased the band gap so that the band gap of Ga$_2$Zr$_{1.9}$W$_{0.1}$O$_7$ (1.81 eV) and Ga$_2$Zr$_{1.85}$W$_{0.15}$O$_7$ (1.7 eV) samples were found to be smaller than that of pyrochlore Sm$_2$Zr$_2$O$_7$ (2.86 eV) and Nd$_2$Zr$_2$O$_7$ (2.67 eV), while Ga$_2$Zr$_{1.8}$W$_{0.2}$O$_7$ has band gap matched with Nd$_2$Zr$_2$O$_7$ (2.67 eV). Full degradation for the CV dye (at 300 min, 25 mmol l$^{-1}$ H$_2$O$_2$) is reached for Ga$_2$Zr$_{1.85}$W$_{0.15}$O$_7$ sample with 15 mol% W doping while lower removal was observed for 20 mol% W-doped sample which is in accordance with their band gaps obtained by DRS as well as the amount of free radical obtained from EPR analysis.

The CV dye photocatalytic degradation followed the pseudo-first-order kinetics. UV–visible and GC–MS studies were conducted to identify the by-products at the end of the reaction of decolourization. GC–MS study indicated that the degradation processes might include $N$-de-methylation followed by aromatic ring rupture. $Ga_2Zr_{1.85}W_{0.15}O_7$ can be used as a promising photocatalyst to purify recalcitrant complicated structure dye for textile water decontamination.

Data accessibility. Data are available from the Dryad Digital Repository: https://doi.org/10.5061/dryad.zpc866t54 [88].
Authors' contributions. A.A.B. and R.A.-Z. carried out the preparation of nano materials, H.A.A. carried out the characterization of the prepared material, T.S.J. and R.A.N. carried out applications of the prepared materials for photocatalytic degradation of crystal violet dye as model compound of textile wastewater. H.A.A. and R.A.N. wrote the manuscript, and T.S.J. and A.A.B. critically revised the manuscript. R.A.N. submitted it.
Competing interests. We declare we have no competing interests.
Funding. This work was funded by the Science and Technology Development Fund in Egypt and the Scientific Research Support Fund in Jordan for financing that work through collaborative project no. 21734 in Egypt and no. Egy-Jor/1/01/2015 in Jordan.
Acknowledgements. Authors acknowledge the Science and Technology Development Fund in Egypt and the Scientific Research Support Fund in Jordan for financing that work through collaborative project no. 21734 in Egypt and no. Egy-Jor/1/01/2015 in Jordan.

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
