## [Reviewer comments · Royal Society Open Science]

Review History

RSOS-191632.R0 (Original submission)

Review form: Reviewer 1

Is the manuscript scientifically sound in its present form?

Yes

Are the interpretations and conclusions justified by the results?

Yes

Is the language acceptable?

Yes

Do you have any ethical concerns with this paper?

No

Have you any concerns about statistical analyses in this paper?

Yes

Recommendation?

Major revision is needed (please make suggestions in comments)

Comments to the Author(s)

In this paper, fluorite type Zr based oxides with the composition $\text{Ga}_2\text{Zr}_{2-x}\text{W}_x\text{O}_7$ ($x=0, 0.05, 0.1$ and 0.15) were prepared using the citrate technique. Appropriate characterizations for all prepared materials were carried out. The Photocatalytic impacts of the prepared systems were studied by removal of crystal violet dye (CV) employing visible illumination taking in consideration initial dye concentrations, duration of visible irradiation treatment, catalysts dose and the dopant concentration. However, there are some questions and suggestions. I suggest it can be considered after revisions.

- (1) Some tests, such as TEM, XPS, and so on, of the as-prepared samples should be tested and discussed.
- (2) It should be given more discussions about the effect of W doping on the photocatalytic performance.
- (3) It should be given more discussions about the W dopant concentration more than 0.15.

Review form: Reviewer 2

Is the manuscript scientifically sound in its present form?

Yes

Are the interpretations and conclusions justified by the results?

Yes

Is the language acceptable?

Yes

Do you have any ethical concerns with this paper?

No

Have you any concerns about statistical analyses in this paper?

No

Recommendation?

Accept with minor revision (please list in comments)

Comments to the Author(s)

The manuscript shows a nano-sized $\text{Ga}_2\text{Zr}_{2-x}\text{W}_x\text{O}_7$ photocatalyst prepared by the Pechini method. The material is prepared successfully in the cubic fluorite phase. As a promising catalyst, it also presents high photocatalytic activity and stability in the progress of removing crystal violet dye. The lowest band gap (1.7 eV) and the highest surface area determine the high efficiency. Therefore, I recommend that the manuscript is considered for publication in Royal Society Open Science after a minor revision. My comments are as followings:

1. Based on other people's results, $\text{O}_2^{\cdot-}$ and HO^{\cdot} are considered to be the active species in the decolorization of CV in haste. But further investigation needs to be done, such as electron paramagnetic resonance (EPR). At the end of abstract, instead of mentioning by-productions, you should talk more about the primary reactive species in the degradation.
2. In Fig. 6, the brackets are not presented in a proper form.
3. The survey XPS spectrum is needed to demonstrate the chemical composition difference in the products.
4. The aesthetics of plotting results needs to be improved.
5. If you draw an energy level diagram of $\text{Ga}_2\text{Zr}_{2-x}\text{W}_x\text{O}_7$, and its corresponding charge separation towards dye degradation under visible light illumination, this will make the manuscript more visual.

6. To investigate the main active species on the catalyst, trapping agents can contrast the different photocatalytic performance of $\text{Ga}_2\text{Zr}_{2-x}\text{W}_x\text{O}_7$ for the degradation of CV with or without adding isopropyl alcohol (IPA), or benzoquinone (BQ), under visible light irradiation.

7. Some references on material preparation and AOPs should be cited, such as *Environmental Science & Technology* 2018, 52, (19), 11297-11308; *Chem* 2019, 5, 2195; *Environmental Science & Technology* 2019, 53, 9725-9733 etc.

Decision letter (RSOS-191632.R0)

06-Jan-2020

Dear Dr Nasr:

Title: Decolorization of crystal violet using nano-sized novel fluorite structure $\text{Ga}_2\text{Zr}_{2-x}\text{W}_x\text{O}_7$ photocatalyst under visible light irradiation

Manuscript ID: RSOS-191632

The editor assigned to your manuscript has now received comments from reviewers. We would like you to revise your paper in accordance with the referee and Subject Editor suggestions which can be found below (not including confidential reports to the Editor). Please note this decision does not guarantee eventual acceptance.

Please submit your revised paper before 29-Jan-2020. Please note that the revision deadline will expire at 00.00am on this date. If we do not hear from you within this time then it will be assumed that the paper has been withdrawn. In exceptional circumstances, extensions may be possible if agreed with the Editorial Office in advance. We do not allow multiple rounds of revision so we urge you to make every effort to fully address all of the comments at this stage. If deemed necessary by the Editors, your manuscript will be sent back to one or more of the original reviewers for assessment. If the original reviewers are not available we may invite new reviewers.

RSC Associate Editor:
Comments to the Author:
(There are no comments.)

RSC Subject Editor:
Comments to the Author:
(There are no comments.)

Reviewers' Comments to Author:
Reviewer: 1

Comments to the Author(s)

In this paper, fluorite type Zr based oxides with the composition $\text{Ga}_2\text{Zr}_{2-x}\text{W}_x\text{O}_7$ ($x=0, 0.05, 0.1$ and 0.15) were prepared using the citrate technique. Appropriate characterizations for all prepared materials were carried out. The Photocatalytic impacts of the prepared systems were studied by removal of crystal violet dye (CV) employing visible illumination taking in consideration initial dye concentrations, duration of visible irradiation treatment, catalysts dose and the dopant concentration. However, there are some questions and suggestions. I suggest it can be considered after revisions.

- (1) Some tests, such as TEM, XPS, and so on, of the as-prepared samples should be tested and discussed.
- (2) It should be given more discussions about the effect of W doping on the photocatalytic performance.
- (3) It should be given more discussions about the W dopant concentration more than 0.15.

Reviewer: 2

Comments to the Author(s)

The manuscript shows a nano-sized $\text{Ga}_2\text{Zr}_{2-x}\text{W}_x\text{O}_7$ photocatalyst prepared by the Pechini method. The material is prepared successfully in the cubic fluorite phase. As a promising catalyst, it also presents high photocatalytic activity and stability in the progress of removing crystal violet dye. The lowest band gap (1.7 eV) and the highest surface area determine the high efficiency. Therefore, I recommend that the manuscript is considered for publication in Royal Society Open Science after a minor revision. My comments are as followings:

1. Based on other people's results, $\text{O}_2\text{-}\bullet$ and $\text{HO}\bullet$ are considered to be the active species in the decolorization of CV in haste. But further investigation needs to be done, such as electron paramagnetic resonance (EPR). At the end of abstract, instead of mentioning by-productions, you should talk more about the primary reactive species in the degradation.
2. In Fig. 6, the brackets are not presented in a proper form.
3. The survey XPS spectrum is needed to demonstrate the chemical composition difference in the products.

4. The aesthetics of plotting results needs to be improved.
5. If you draw an energy level diagram of $\text{Ga}_2\text{Zr}_{2-x}\text{W}_x\text{O}_7$, and its corresponding charge separation towards dye degradation under visible light illumination, this will make the manuscript more visual.
6. To investigate the main active species on the catalyst, trapping agents can contrast the different photocatalytic performance of $\text{Ga}_2\text{Zr}_{2-x}\text{W}_x\text{O}_7$ for the degradation of CV with or without adding isopropyl alcohol (IPA), or benzoquinone (BQ), under visible light irradiation.
7. Some references on material preparation and AOPs should be cited, such as Environmental Science & Technology 2018, 52, (19), 11297-11308; Chem 2019, 5, 2195; Environmental Science & Technology 2019, 53, 9725-9733 etc.

Author's Response to Decision Letter for (RSOS-191632.R0)

See Appendix A.

RSOS-191632.R1 (Revision)

Review form: Reviewer 1

Is the manuscript scientifically sound in its present form?

Yes

Are the interpretations and conclusions justified by the results?

Yes

Is the language acceptable?

Yes

Do you have any ethical concerns with this paper?

No

Have you any concerns about statistical analyses in this paper?

No

Recommendation?

Accept as is

Comments to the Author(s)

The authors have addressed the significant points needed to be revised. I suggest this paper published in Royal Society Open Science.

Decision letter (RSOS-191632.R1)

11-Feb-2020

Dear Dr Nasr:

Title: Decolorization of crystal violet using nano-sized novel fluorite structure $\text{Ga}_2\text{Zr}_{2-x}\text{W}_x\text{O}_7$ photocatalyst under visible light irradiation
Manuscript ID: RSOS-191632.R1

It is a pleasure to accept your manuscript in its current form for publication in Royal Society Open Science. The chemistry content of Royal Society Open Science is published in collaboration with the Royal Society of Chemistry.

RSC Associate Editor:
Comments to the Author:
(There are no comments.)

RSC Subject Editor:
Comments to the Author:
(There are no comments.)

Reviewer(s)' Comments to Author:
Reviewer: 1

Comments to the Author(s)
The authors have addressed the significant points needed to be revised. I suggest this paper published in Royal Society Open Science.

Appendix A

Dear Valuable editor of the valuable Royal Society Open Science

Dear valuable and distinguishable reviewers.

Authors would like to thank you all for your effort in handling, reviewing and great inputs that really improve the quality of our manuscript.

Authors went through the comments and replied to all comments and did amended the manuscript according to the comments of our valuable reviewers. Authors hope that the manuscript in its current form is qualified to be published in the valuable Royal Society Open Science. All the corrected parts are highlighted in green.

Here is in the next table you can find the reply of all comments as well as its place in the manuscript

Comments	Reply
Reviewer #1	
Some tests, such as TEM, XPS, and so on, of the as-prepared samples should be tested and discussed.	Thank you for the comment. TEM is presented in the supplementary S4. IR spectra were recorded for all the samples and presented in Fig. 3. Electron paramagnetic Resonance (EPR) was performed for all the samples (Fig. 6 and Table 3). In Egypt there is only one XPS equipment which is unfortunately is down in this month and we waited for the whole time to be maintained but it does not work. As well, authors proved the chemical composition of the prepared samples by ICP/OES (Table 1) to determine the practical amount of Ga, Zr and W and compare it with the theoretical values that has been theoretically determined before the preparation. Moreover, the EPR (Fig 6 And Table 3) indicated presence of Zr^{+3} as a result of W doping.
It should be given more discussions about the effect of W doping on the photocatalytic performance.	Thank you for the comment The effect of W doping on the photocatalytic performance was manifested at section 3.2.4
It should be given more discussions about the W dopant concentration more than 0.15	Thanks for the comment A sample containing $x=0.2$ ($GA_2Zr_{1.8}W_{0.2}O_7$) was prepared, fully characterized and introduced to the necessary parts in the manuscript. The

	discussion was amended to align with the presence of the new dopant concentration.
Reviewer #2	
Based on other people's results, $O_2^{\bullet-}$ and HO^{\bullet} are considered to be the active species in the decolonization of CV in haste. But further investigation needs to be done, such as electron paramagnetic resonance (EPR). At the end of abstract, instead of mentioning by-productions, you should talk more about the primary reactive species in the degradation.	Thanks for the comment EPR was performed as required by the valuable reviewer (Section 3.1). The reactive species experiment was performed by introducing scavenger effect experiment to the manuscript (section 3.2.9.) and then the abstract is modified.
In Fig. 6, the brackets are not presented in a proper form.	Thank you for the comment The caption of Fig 6 has been amended taking into account that the effect of time experiment was carried out before optimizing dose or pH.
The survey XPS spectrum is needed to demonstrate the chemical composition difference in the products.	Thank you for the comment. In Egypt there is only one XPS equipment which is unfortunately is down in this month and we waited for the whole time to be maintained but it does not work. As well, authors proved the chemical composition of the prepared samples by ICP/OES (Table 1) to determine the practical amount of Ga, Zr and W and compare it with the theoretical values that has been theoretically determined before the preparation. Moreover, authors performed EPR (Fig 6 And Table 3) that indicates presence of Zr^{+3} as a result of W doping.
The aesthetics of plotting results needs to be improved	Thank you for the comment, authors aesthetic the plotting as much as possible and we hope that the results seems better now.
If you draw an energy level diagram of $Ga_2Zr_{2-x}W_xO_7$, and its corresponding charge separation towards dye degradation under visible light illumination, this will make the manuscript more visual.	Thank you for the comment. Authors amended the drawn energy level diagram (Supplementary information, S3) of $Ga_2Zr_{2-x}W_xO_7$ to show its corresponding charge separation towards dye degradation under visible light illumination.
. To investigate the main active species on the catalyst, trapping agents can contrast the different photocatalytic performance of $Ga_2Zr_{2-x}W_xO_7$ for the	Thank you for the comment. To evaluate of active species, three scavengers were used (Isopropyl Alcohol,

degradation of CV with or without adding isopropyl alcohol (IPA), or benzoquinone (BQ), under visible light irradiation.	Ethylene diamine tetra acetic acid (EDTA) and Benzoquinone) for HO^\bullet, h^+ and $\text{O}_2^{\bullet-}$ species respectively (section 3.2.9) indicating that h^+ and $\text{O}_2^{\bullet-}$ had the major contribution in photocatalytic degradation of CV while, HO^\bullet had a minor contribution.
. Some references on material preparation and AOPs should be cited, such as Environmental Science & Technology 2018, 52, (19), 11297-11308; Chem 2019, 5, 2195; Environmental Science & Technology 2019, 53, 9725-9733 etc	Thank you for the comment The references are cited as required by the valuable reviewer

Authors would like to thank the reviewers once again for the efforts in correcting our manuscript

**Best wishes and regards
Authors**